# Music and low-frequency vibrations for the treatment of chronic musculoskeletal pain in elderly: A pilot study

Thom A. H. Eshuis[1]☯*, Peter J. C. Stuijt[1,2]☯, Hans Timmerman[1,3], Peter Michael L. Nielsen[4], André Paul Wolff[1], Remko Soer[1,3,5]

**1** Department of Anaesthesiology, University of Groningen, University Medical Center Groningen, UMCG Pain Center, Groningen, The Netherlands, **2** Department of Human Movement Sciences, University of Groningen, University Medical Center Groningen, Groningen, The Netherlands, **3** Department of Rehabilitation Medicine, University of Groningen, University Medical Center Groningen, Groningen, The Netherlands, **4** Department of Neurology, Holbaek Hospital, Part of Copenhagen University Hospital, Sjaelland, Denmark, **5** Saxion University of Applied Sciences, Expertise Center of Health and Movement, Enschede, The Netherlands

☯ These authors contributed equally to this work.
\* t.a.h.eshuis@umcg.nl

**Data Availability Statement:** All relevant data are within the paper and its Supporting Information files.

## Abstract

### Background

Transcutaneous vagal nerve stimulation has analgesic potential and might be elicited by abdominally administered low-frequency vibrations. The objective was to study the safety and effect of a combination of music and abdominally administered low-frequency vibrations on pain intensity in elderly patients with chronic musculoskeletal pain.

### Methods

This trial was an international multicenter, randomized controlled pilot study. Patients at age $\geq$ 65 years with musculoskeletal pain for $\geq$ 3 months and a daily pain score $\geq$ 4 out of 10 were recruited at three centers. They were randomized to receive either a combination of music and low-frequency (20–100 Hz) vibrations administered to the abdomen, or a combination with the same music but with higher frequency (200–300 Hz) vibrations administered to the abdomen. Low-frequency vibrations were expected to result in pain reduction measured with a numeric pain rating scale (NRS). Patients in both groups received eight treatments of the music combined with the vibrations in three weeks. Primary outcomes were safety (Serious Adverse Events) and pain intensity measured at baseline, after the last treatment and at six weeks follow-up. Multilevel linear model analyses were performed to study group and time effects.

### Results

A total of 45 patients were analyzed according to intention-to-treat principle. After 344 treatments, 1 Adverse Event was found related to the intervention, while 13 Adverse Events were possibly related. A multilevel linear model showed that the interaction effect of group

**Funding:** RS was in 2018 granted by the Active Assisted Living Programme (AAL EU). Grant number: AAL2018-5-144-SCP URL: https://www.aal-europe.eu/ The funder had no role in study design, data collection and analysis, decision to publish, or preparation of the manuscript.

**Competing interests:** Peter Michael Nielsen is registered inventor of the HALF-MIS technology. As a consequence, Peter Michael Nielsen did not participate in statistics processing or discussion of the results. This does not alter our adherence to PLOS ONE policies on sharing data and materials. For the remaining authors, none were declared.

by time did not predict pain intensity ($F$[1, 45.93] = 0.002, p = 0.97) when comparing pain intensity at baseline, after the last treatment and at follow-up.

## Conclusions

The combination of music and abdominally administered vibrations was found to be safe and well tolerated by the elderly patients. However, over time, neither the low-frequency treatment group nor the high-frequency treatment group provided clinically meaningful pain relief. There is no evidence that the low-frequency treatment elicited vagal nerve stimulation.

## Trial registration

The trial was prospectively registered in the Netherlands Trial Register (NTR: NL7606) on 21-03-2019.

## Introduction

When looking into the impact of chronic musculoskeletal pain (CMP), it is reported that low back pain, neck pain and pain related to osteoarthritis are respectively the leading, 4th and 13th biggest causes of global years lived with disability, with an accumulated mean total of over 119 million in 2013 [1, 2]. Yet, the years lived with disability caused by these conditions are rising due to the aging of the global population, illustrating the enormous challenge that pain management is facing [3–7].

Most commonly reported treatments for CMP include the intake of drugs (e.g. paracetamol, NSAID's and/or opioid analgesics [4]) and non-drug treatments such as physical- and exercise therapies, and transcutaneous electrical nerve stimulation [4, 8–10]. Interestingly, 31% of patients with CMP report not to be treated anymore for various reasons, including but not limited to not experiencing any effect of therapy on pain and suffering from side effects such as constipation and nausea [4]. The latter applies especially to elderly, as ageing is generally associated with higher prevalence of comorbidities and polypharmacy [11], which reduces appropriate pharmacological and invasive treatment options.

A relatively new approach in chronic pain management, is vagal nerve stimulation (VNS) [12]. Effects of VNS have been reported to alleviate pain, depression, anxiety, migraine and refractory epilepsy [12, 13]. Although it is unknown how VNS exactly results in relief in the aforementioned conditions, findings show that stimulation of vagal afferents elicits anti-inflammatory responses via activation of the cholinergic anti-inflammatory pathway [14] and potentially suppresses the transmission of pain signals both in the peripheral and central nervous system [15, 16]. VNS is mostly administered by a surgically implanted pulse generator that stimulates the left vagal nerve rostral of the aortic arch [12]. Downsides of this invasive method are the risk of side effects caused by efferent stimulation such as bradycardia, arrhythmia, cough and voice alteration and complications due to the surgery (e.g. infections and vocal cord paresis) [12, 13, 17, 18]. Non-invasive transcutaneous VNS (t-VNS) has already been proven safe and effective in the treatment of migraine, epilepsy, depression and cluster headache [19–22]. However, the results of studies examining the effect of t-VNS on the perception of pain have been inconsistent [23, 24]. Currently, t-VNS has mainly been attempted by

electrically stimulating either auricular or cervical branches of the vagal nerve, but recent preliminary results suggest that VNS might also be induced by vibratory stimulation [21].

While vibratory stimulation has already proven to be effective in alleviating chronic pain [25–29], the effect has mainly been ascribed to the mechanisms of gate control theory [30]. According to this theory, the transmission of nociceptor signaling via small diameter C fibers can be inhibited in the dorsal horn by simultaneous stimulation of local mechanoreceptors with large diameter Aβ fibers [30, 31]. It is hypothesized that exposing the abdomen to vibratory stimulation could result in VNS; a considerable amount of vagal afferents innervate abdominal organs including the gastrointestinal tract, the liver and the pancreas [13, 32] and an abundance of mesentery nerves are thought to innervate Pacinian corpuscles [33]. These rapidly adapting corpuscles are known to be highly sensitive to pressure and vibrations with frequencies ranging from 20 to 1000 Hz [33], and send afferent stimuli through myelinated αβ fibers via the thalamus to SI and SII in the cortex [34]. Although the abdominal vagal afferents and the mesenterially innervated Pacinian corpuscles have not been linked in anatomical studies with one another, there are studies indicating a relationship between these two [35–39]. For instance, it is found that the combination of music and vibration that is used in vibro-acoustic therapy evokes changes in pain perception, heart rate variability, respiration rate, endocrine function [36, 37], which are all regulated by the vagal nerve [35, 40]. Moreover, it is observed that during vibro-acoustic therapy especially lower frequency vibrations (ranging from 30 to 120 Hz) were more effective in abdominal pain relief than higher frequency vibrations [37, 38]. This supports the idea that application of low-frequency vibration in the mesentery might result in VNS, as lower frequency vibrations have the ability to travel further through the skin than higher frequency vibrations with the same sound pressure [39].

T-VNS is a promising treatment, especially for patients that suffer from side-effects of medication or are not fit for invasive treatment due to comorbidities. Presumptively, t-VNS can be achieved by exposing the abdomen to low frequency vibrations, and combining it with music increases its analgesic potential [41–43]. This pilot study aimed to examine the safety and effect of music and low-frequency vibrations administered to the abdomen on CMP in elderly patients in addition to their current pain treatment(s). Both combinations of music and vibrations were expected to be safe and well tolerated by the elderly patients with CMP. It was also hypothesized that (I) the combination of vibrations administered to the abdomen and rhythmically aligned music would have a clinically meaningful analgesic effect in elderly with CMP and (II) the combination of music and vibrations with frequencies between 20–100 Hz would be more effective in relieving CMP in elderly than the combination of music and vibrations with higher frequencies (200–300 Hz), as in contrast to higher frequency vibrations, lower frequency vibrations could result in t-VNS.

## Materials and methods

### Study design

This study was an international multicenter, randomized, double-blind, pilot trial conducted between August 2019 and March 2020. The three centers that participated in this trial were Holbæk Hospital (Holbæk, Denmark), San Martino Hospital (Genova, Italy) and University Medical Center Groningen (Groningen, the Netherlands). The study duration per patient was 9 weeks. The medical ethics committee of UMC Groningen had approved the study (2019/227;NL69608.042.19), in accordance with the Helsinki declaration. The study protocol is available upon request. Written informed consent was obtained in advance from each patient. The trial was registered in the Netherlands Trial Register (NTR: NL7606) prior to the inclusion of the first patient.

## Patients

Patients suffering from CMP were targeted by means of flyers hung in public places, media coverage of the trial, and by consultation of local general practitioners and in-clinic medical specialists. Interested patients were pre-screened during phone calls for suitability and if deemed suitable scheduled for examination to assess study eligibility definitely. Patients were included when at least 65 years, and suffering from symptoms of pain for at least 3 months, with a minimum intensity of Numeric (Pain) Rating Scale (NRS) of 4. Moreover, the pain had to be the result of a condition diagnosed as a musculoskeletal disease listed in the International Classification of Diseases-10 of 2014 under M00-M99 [44]. Patients participating in other (experimental) clinical studies, undergoing any kind of music therapy as a current treatment for their pain, suffering from active or untreated comorbidities (including moderate to severe depression), having pain related to malignancies, showing signs of the pain syndrome being exclusively or predominantly neurogenic during physical examination or being diagnosed with prolapsus disci intervertebralis with myelopathy/radiculopathies (listed as M50.0, M50.1, M51.0 and M51.1 in the International Classification of Diseases-10 of 2014) were excluded [44]. Patients were allowed to continue or even change both their pharmacological and non-pharmacological treatments (if applicable), but were requested to report any changes in treatment.

## Procedures

Participants were randomized 1:1 per center to either regular 'High Amplitude Low Frequency–Music Impulse Stimulation' (HALF-MIS) treatment (i.e. the low-frequency group) or a variation of the HALF-MIS treatment, in which the low-frequency vibrations (20–100 Hz) of the regular HALF-MIS had been replaced by higher-frequency vibrations (200–300 Hz). This last treatment was considered to be an active control group for t-VNS, as higher frequency vibrations would not have the potential to result in vagal nerve stimulation. This would allow us to evaluate the second hypothesis, by purely examining the analgesic effect of t-VNS in addition to the effects music, rest, placebo and the mechanism of the gate control theory of pain on CMP in elderly. Before the inclusion of the first patient, a random sequence was created by T.E. and P.S. by matching study-ID's to handpicked allocations. Then, for every future study-ID, a sealed opaque envelope containing the blinded group allocation ("A" or "B") was distributed to the centers. The envelope was to be opened after a patient gave written informed consent and was found fit for inclusion. To ensure blinding of the patient, the equipment used for both treatment groups was also marked with "A" or "B", but looked identical. Allocation was concealed for the patients until data collection was completed and for the primary researcher (RS) until the results were analyzed.

The San Martino Hospital suffered during the intervention period of the trial from both flooding and the inability to operate one of two treatment devices. These unfortunate circumstances led to a situation, in which protocol violations could not be averted. After the monitoring of all data by an independent monitor, it was concluded that the protocol violations were too severe for the data to be included in the analyses. That is the reason why the data of this center is not included in the CONSORT flow diagram (see Fig 1).

## Interventions

Treatments were administered by physicians T.E. and P.N., in the Netherlands and Denmark, respectively. During a treatment, patients sat or lied in the chair while being exposed to rhythmically aligned music administered through a headphone and synchronized vibrations, via a belt worn around the waist (see Fig 2). Both groups received 8 treatments of 24 minutes and 7

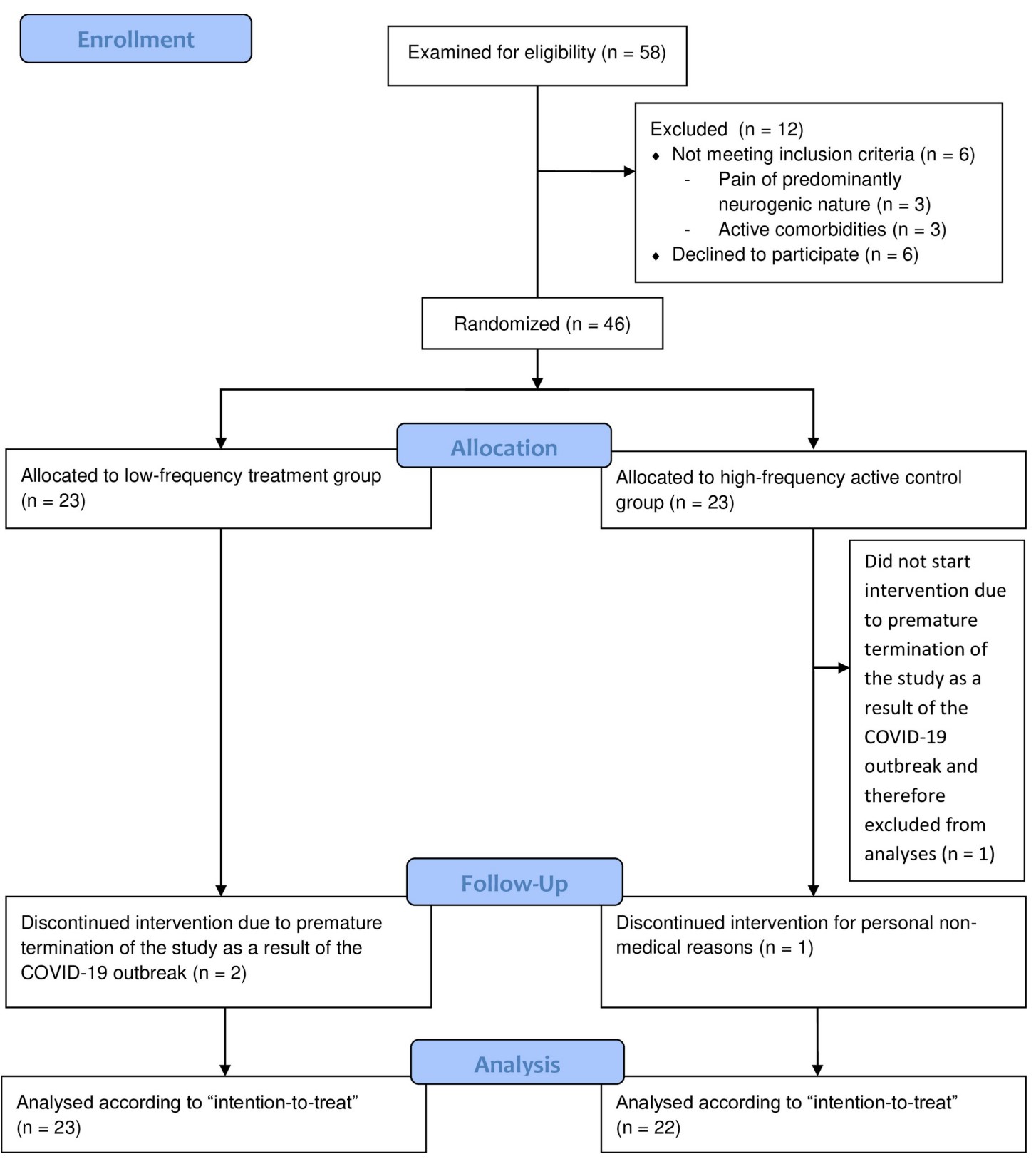

**Fig 1. The CONSORT flow diagram.**

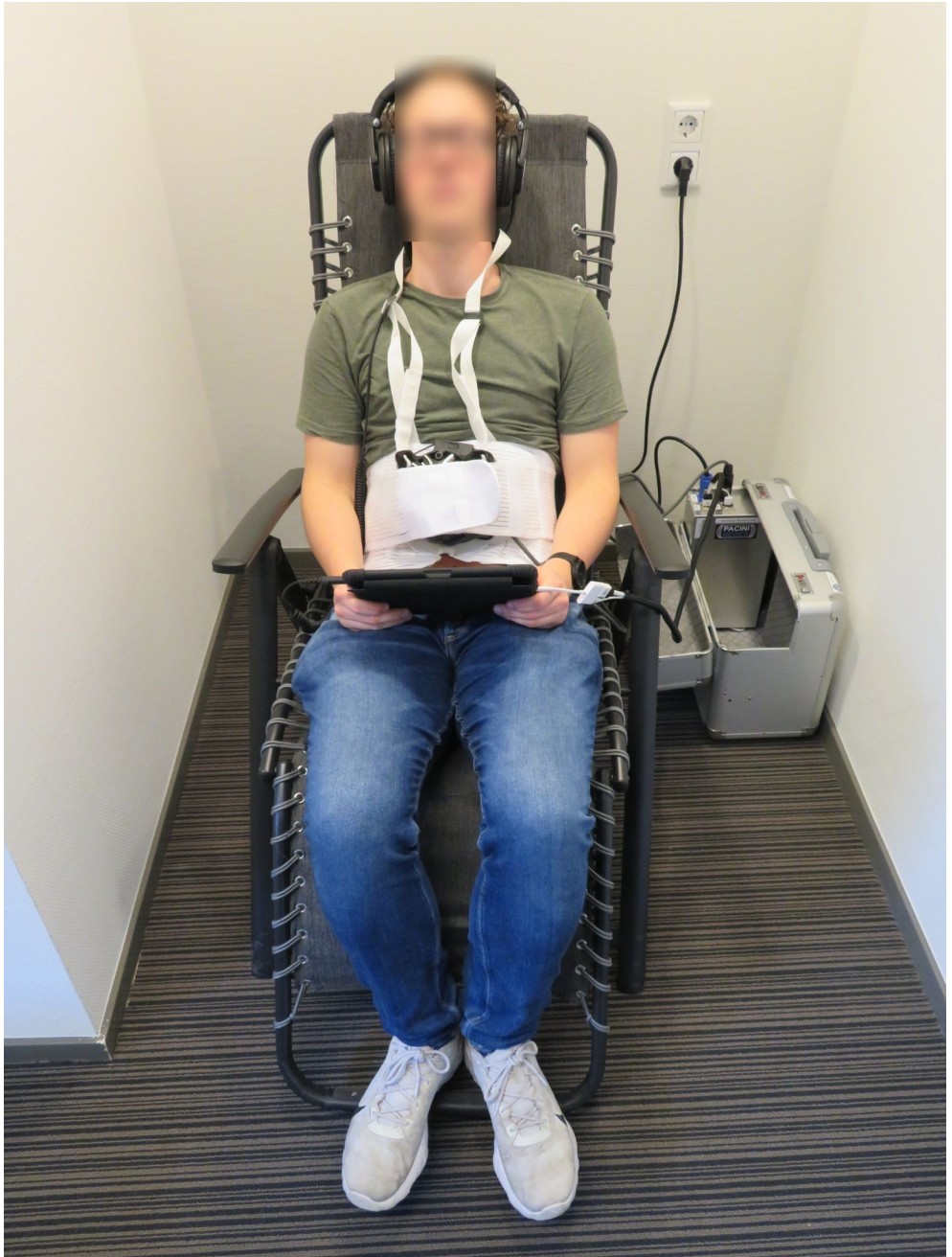

**Fig 2. The equipment used for both the HALF-MIS treatment and the higher frequency variant in a clinical setting.** Reprinted from T.A.H. Eshuis et al. under a CC BY license, with permission from T.A.H. Eshuis et al., original copyright [2021].

seconds (corresponding with the duration of the music) within three weeks, with at least one day without treatment between two consecutive treatments [21]. A follow-up visit was conducted 6 weeks after the last treatment. The equipment used for the HALF-MIS treatment was provided by PaciniMedico ApS (Copenhagen, Denmark). This consisted of a comfortable and adjustable chair, an elastic belt with a low frequency audio transducer (Guitammer, Westerville, OH), headphones, a central processing unit and a tablet (Apple Inc., Cupertino, CA) for

the user interface on which both music volume and vibration intensity could be modified by the patient. The room in which the treatments took place, was standardized as much as possible throughout the intervention period to minimize exposure to external stimuli that could affect pain processing; windows were always closed, blinds were always down and the room was free of unnecessary material.

## Outcomes

The primary outcome to assess the safety of the intervention was the quantity and quality of reported (Serious) Adverse Events that were possibly related to the intervention([S]AE's). The primary outcome to assess the efficacy of both treatments was patient perceived pain intensity score. The NRS score provides a valid and reliable measure for pain intensity for this study population, and measures pain on a 11-point scale ranging from 0 (no pain) to 10 (the worst pain imaginable) [45]. The effects of both treatments were assessed by a comparison of the pain intensity measured before the first treatment (baseline), after the last treatment and at follow-up.

The secondary outcomes were immediate pain relief, scores of the health-related quality of life (EQ-5D-3L), and pain disability (Pain disability Index; PDI). Immediate pain relief was quantified as the difference between the NRS score obtained immediately before and after each treatment, respectively and assessed for both groups. The health-related quality of life and pain disability questionnaires were filled out before the start of the first and eighth treatment and at follow-up, while screening for possible (S)AE's took place at each visit. The EQ-5D-3L has been proven a valid tool for assessing quality of life in Danish, Dutch and Italian [46] and is found to be reliable when used in a Dutch elderly population [47]. The EQ-5D-3L is a self-report questionnaire that consists of 5 health dimensions (mobility, self-care, usual activities, pain/discomfort and anxiety/depression), which are all scored on a 3-point scale. It has a minimum score of -0.33 (no perceived quality of life) and a maximum score of 1 (highest quality of life possible perceived). Per country, the appropriate EQ-5D-3L valuation set was used for calculation of the total score, to account for sociocultural differences with regard to quality of life perspectives [48, 49]. The PDI has been proven a valid and reliable tool for measuring disability associated with pain in Dutch, and consists of 7 items which are scored on a 11-point scale [50]. The total score is calculated by the sum of scored item scores, divided by the number of scored items [51]. This yields a percentage ranging from 0 (no disability experienced) to 100 (experience of total disability). For this study, the PDI has been translated using a formal forward-backward method into Italian and Danish. To examine the course of quality of life and pain disability over time for each treatment group, within-group differences were compared for before the first treatment, after the last treatment and at follow-up.

Use of analgesics over time during intervention period was considered a possible confounder and was therefore also recorded at the screening visit. Moreover, changes in analgesic use were kept track of at each visit. Analgesics intake was then quantified using the Analgesics Intake Scale [52]. This tool yields a score from 0 (no use of pain medication) to 8 (use of opioids in combination with benzodiazepines/antidepressant/anticonvulsants), and allows for comparing analgesic intake between patients by taking into account both type and quantity of medication used.

## Sample size

Since this was the first trial investigating the effect of HALF-MIS on CMP in elderly, there was no published or unpublished pilot data available to base a sample size on. For this pilot study, it was decided to aim for inclusion of 60 patients in total (i.e. 30 patients per treatment group).

## Statistical analysis

All analyses are performed according the intention-to-treat principle to avoid potential exclusion bias, meaning that patients who started but did not finish the intervention were included in the analyses [53]. Differences in age, sex distribution, weight, height and body mass index between groups were compared using tests depending on data distribution and level of measurement.

The effects of various possible predictors on pain intensity were analyzed by means of a multilevel linear model to examine the effect of both treatments over a longer period of time, both within and between treatment groups. Heterogeneous first-order autoregressive structure was used as repeated covariance type. The model was built by starting with a fixed intercept and the interaction effect of group by time, as this was the effect of interest to check the second hypothesis. The model was then expanded with factors (both fixed and random) that significantly improved the model fit ($\chi^2$ [1] > 3.84) [54]. As it was expected that the effect of time would not be linear, different ways of coding the three most meaningful moments in time (i.e. before the first treatment, after the last treatment and at follow-up, respectively) had been included. As multilevel models do not require completeness of data, missing data was not imputed.

To assess the immediate pain relief within treatment groups, the average NRS score per patient before the delivered treatments was compared to the average NRS score per patient after the delivered treatments for both groups using Wilcoxon signed-rank test. The average difference between pre- and post-treatment NRS per patient between treatment groups were compared by means of Mann-Whitney test. Exclusion bias and missing data bias have been reduced by first taking the mean of all pre-treatment and post-treatment NRS scores for the Wilcoxon signed-rank test and mean difference per patient between each pre- and post-treatment NRS scores for the Mann-Whitney test, before conducting the tests mentioned. Effect sizes for Wilcoxon signed-rank test and Mann-Whitney test were calculated as $r$ [54]. A $r$-value < 0.3 represents a small effect, $0.3 \leq r \leq 0.5$ represents a medium-size effect and $r > 0.5$ represents a large effect [54]. For the remaining secondary outcomes quality of life and pain disability, changes within groups were examined by a comparison of within-group differences at baseline, after the last treatment and at follow-up. This was done by means of Friedman's two-way ANOVA's as subgroups of both variables were not normally distributed judged by visual inspection. To control for familywise error in the comparison of within-group differences for the secondary outcomes, Bonferroni correction was used and the two-sided level of statistical significance was set to 0.0125. All other statistical analyses were conducted with a two-sided 0.05 level of statistical significance, using SPSS statistics (version 23.0; IBM Corp. Armonk, NY).

## Results

A total of 46 patients have been included and randomized (see Fig 1 and Table 1). However, one patient had not started the intervention when the study was terminated due to the COVID-19 outbreak, and was therefore excluded from the analyses. Three patients have not completed the intervention period and missed a total of 16 treatments (and accessory pain intensity measures); these were 9 low-frequency and 7 high-frequency treatments. Consequently, the 45 randomized patients have received a total of 344 treatments, during which 686 pain scores were obtained. In each group, 1 pain score was missing. The total number of follow-up visits that were performed is 42; 21 for each group. Of these visits, no data was missing. One non-pharmacological treatment during the intervention period was reported; a patient visited her physiotherapist. Pain intensity scores along with analgesic use at baseline, after the

**Table 1. Descriptive characteristics per group at baseline (Mean ± SD or Median [IQR], except for sex).**

| | Low-frequency group (n = 23) | High-frequency group (n = 22) | p-value[1] |
|---|---|---|---|
| **Age (years)** (Median [IQR]) | 72.0 [68.0–77.0] | 73.0 [66.0–79.3] | 0.65 |
| **Sex (M/F)** (Ratio) | 6/17 | 6/16 | 0.98 |
| **Weight (kg)** (Median [IQR]) | 78.5 [68.8–95.3] | 72.0 [61.0–78.0] | 0.08 |
| **Height (cm)** (Mean ± SD) | 165.8 ± 9.9 | 164.7 ± 9.0 | 0.69 |
| **Body mass index (kg/m²)** (Median [IQR]) | 28.9 [24.9–33.9] | 25.9 [23.4–28.6] | 0.08 |

[1]p-value of either Mann-Whitney test (age, body mass index and weight), $\chi^2$-test (sex) or independent T-test (height).

last treatment and at follow-up are given in Table 2, while secondary outcomes are given in Table 3.

The treatments appeared well received and safe. No Serious Adverse Events that were possibly related to the intervention were reported. One non-serious Adverse Event was reported that was related to the treatments as one subject perceived low back pain caused by poor posture while treatment was administered. 13 other non-serious Adverse Events might have been related, of which short episodes of respectively dizziness/vertiginous (n = 5) and headache (n = 2) and tiredness (n = 2) have been reported more than once.

With regard to the effectiveness of both treatments, the parameter estimates of the multi-level linear model are shown in Table 4. Independent variables that were examined as possible predictors were age, sex, body mass index, Analgesic Intake scale score, group (0 = high-frequency group, 1 = low-frequency group), time (0 = baseline, 0.5 = after last treatment, 1 = at follow-up) and center (0 = Holbaek, 1 = Groningen). Only center, group, and time significantly contributed as fixed (interaction) effect to the model fit and were therefore included as parameters. The group by time effect did not significantly predict pain intensity, $F(1, 45.93) = 0.002$, p = 0.97. The use of random slopes did not result in any improvement of the model. However, the relationship between the included predictors and pain did show significant variance in intercepts across patients of both groups, $Var(u_{0j}) = 3.61$, $\chi2(1) = 17.71$, p < 0.01. Only pain intensity at baseline (i.e. the intercept of the model) and center were significant predictors of pain intensity at later stages, while an interaction of group by time by center was close to being a significant predictor.

With regard to the immediate effects of both treatments, related-samples Wilcoxon Signed Rank Tests showed for both treatment groups that the average NRS score per patient after treatment was significantly lower than the average NRS per patient before treatment: for the

**Table 2. Pain intensity and analgesic use per group and for total sample before the first treatment (T0), after the last treatment (T1) and at follow-up (T2) (Mean ± SD or Median [IQR]).**

| | T0 | T1 | T2 |
|---|---|---|---|
| **NRS (Mean ± SD)** | | | |
| Low-frequency group | 5.4 ± 2.5 | 5.1 ± 2.3 | 5.0 ± 2.6 |
| High-frequency group | 5.4 ± 2.3 | 4.0 ± 2.7 | 5.4 ± 2.1 |
| Total sample | 5.4 ± 2.4 | 4.5 ± 2.5 | 5.2 ± 2.4 |
| **Analgesics Intake Scale (Median [IQR])** | | | |
| Low-frequency group | 2 [0–7] | 2 [1–7] | 2 [1–7] |
| High-frequency group | 2 [0.75–7] | 1 [0–7] | 1 [0–7] |
| Total sample | 2 [0–7] | 2 [0–7] | 2 [0–7] |

[1]p-value of Friedman's ANOVA.

**Table 3. Secondary outcomes per group and for total sample before the first treatment (T0), after the last treatment (T1) and at follow-up (T2) (Median [IQR]).**

|  | T0 | T1 | T2 | p-value[1] |
|---|---|---|---|---|
| **EQ-5D-3L** |  |  |  |  |
| **Low-frequency group** | 0.72 [0.48–0.78] | 0.78 [0.52–0.79] | 0.65 [0.49–0.79] | p = 0.70 |
| **High-frequency group** | 0.65 [0.42–0.81] | 0.70 [0.48–0.81] | 0.67 [0.43–0.81] | p = 0.94 |
| **Total sample** | 0.69 [0.45–0.78] | 0.76 [0.50–0.81] | 0.65 [0.48–0.81] |  |
| **Pain Disability Index** |  |  |  |  |
| **Low-frequency group** | 31.4 [10.0–40.0] | 32.9 [11.5–45.4] | 32.9 [21.7–54.2] | p = 0.09 |
| **High-frequency group** | 37.5 [27.0–44.3] | 30.0 [12.1–48.0] | 42.9 [20.7–56.7] | p = 0.07 |
| **Total sample** | 31.4 [13.3–41.4] | 32.1 [11.6–45.2] | 36.4 [21.6–54.2] |  |

[1]p-value of Friedman's ANOVA.

low-frequency group, the average NRS score after treatment (Mdn = 3.9, IQR = 2.3–5.0) was lower than the average NRS score before treatment (Mdn = 4.8, IQR = 3.4–5.4), p < 0.01, r = -0.49, while for the high-frequency group, the average NRS score after treatment (Mdn = 3.6, IQR = 2.5–5.3) was lower than the average NRS score before treatment (Mdn = 4.3, IQR = 3.3–6.6), p < 0.01, $r$ = -0.50. The average difference per patient between pre- and post-treatment NRS scores from the low-frequency group (Mdn = 0.8, IQR = 0.4–1.5) did not differ significantly from those of the high-frequency group (Mdn = 0.6, IQR = 0.1–1.3), U = 235.00, $z$ = -0.41, p = 0.68, $r$ = -0.06.

Neither for the low-frequency group, $\chi^2(2)$ = 0.71, p = 0.70, nor for the high-frequency group, $\chi^2(2)$ = 0.12, p = 0.94, the scores of the EQ-5D-3L changed over time. Similarly, neither treatment group showed significant changes over time in Pain Disability Index score: $\chi^2(2)$ = 4.84, p = 0.09 for the low-frequency group and $\chi^2(2)$ = 5.29, p = 0.07 for the high-frequency group, respectively.

## Discussion

This pilot study aimed to examine the safety and effect of music and low-frequency vibrations administered to the abdomen on CMP in elderly patients in addition to their current pain treatment. It was hypothesized that the combination of music and vibrations with frequencies between 20–100 Hz would be more effective in relieving CMP in elderly than the combination

**Table 4. Parameter estimates of the fixed effects of the final model predicting pain intensity.**

|  | b[1] | SE$_b$ | 95% CI |
|---|---|---|---|
| Baseline NRS score | 5.57** | 0.47 | 4.62, 6.53 |
| Center | -1.43* | 0.64 | -2.70, -0.15 |
| Group x Time | -0.02 | 0.58 | -1.20, 1.15 |
| Group x Time x Center | -1.46 | 0.81 | -3.09, 0.16 |

Independent variables: center (0 = Holbaek, 1 = Groningen), group (0 = high-frequency group, 1 = low-frequency group) and time (0 = baseline, 0.5 = after last treatment, 1 = at follow-up).

The relationship between group and pain showed significant variance in intercepts across patients, Var($\mu_{0j}$) = 3.61, $\chi^2(1)$ = 17.71, p < 0.01.

\* = p < 0.05.

\*\* = p < 0.01.

[1]b = unstandardized regression coefficient.

of music and vibrations with higher frequencies (200–300 Hz). However, there was no statistically significant interaction effect of group by time in the multilevel linear model predicting pain intensity, meaning that the low-frequency treatment was not more effective than the high-frequency treatment when comparing pain intensity at baseline, after the last treatment and at follow-up, respectively. It was also expected that the combination of vibrations administered to the abdomen and rhythmically aligned music would have a clinically meaningful analgesic effect in elderly with CMP, but as no time effect was found either, it seems that both treatments do not decrease CMP in elderly in the long run. The expectation that the treatment would be safe and well tolerated by the patients was confirmed, judging by both the quantity and quality of reported (Serious) Adverse Events that were possibly related to the intervention.

Patients with a higher baseline pain intensity score and patients treated in Holbæk showed higher pain intensity at a given point in time then patients with a lower baseline pain intensity score or those being treated in Groningen, respectively. The latter can be explained by a difference in recruitment; while most Dutch patients showed interest in the study after reading about the study in a newspaper without being previously referred to the clinic, Danish patients were recruited after consultation of the in-clinic specialists. This second approach has yielded patients with more severe pain syndromes. The interaction effect of group by time by center was close to being a statistically significant predictor of pain intensity. In both Danish and Dutch low-frequency groups, mean pain intensity was decreased non-significantly when comparing the 'after the last treatment' to baseline values but non-significantly increased at follow-up compared to 'after the last treatment'. While this trend is also found in the Dutch high-frequency group, the Danish high-frequency group showed no change in mean pain intensity over time at all.

However, both the low-frequency and the high-frequency treatments showed a statistically significant decrease in pain intensity when pain intensity scores before and after the treatment are compared. While the effect sizes are moderate ($r$ = -0.49) for the low-frequency and large ($r$ = -0.50) for the high-frequency group, these changes are not always significant from the patient's perspective; as the median drop in mean pain intensity score per patient is 19% for the low-frequency group and 16% for the high-frequency group, most patients experience only partial temporary relief of their moderate to severe musculoskeletal pain immediately after treatment. Still, the immediate effects found in this study seem larger than immediate-term effects of oral and transdermal opioid analgesics on pain intensity in elderly with CMP found by a systematic review, which showed a standardized mean difference of 0.27 in a sample of 4998 patients by combining 16 studies [55].

The analgesic effect on musculoskeletal pain of both treatments that is observed immediately after the treatment, can hypothetically arise from the interaction of five components: rest, placebo, music [41–43], the mechanism of the gate control theory of pain [25–29] and VNS [23, 24]. It was hypothesized that only the low-frequency treatment could lead to VNS, as only the low-frequency vibrations were expected to have the ability to reach and stimulate vagal afferents in the mesentery. This would either mean that the low-frequency treatment did not result in VNS, or that VNS did not enhance the analgesic effect of the other component(s).

No changes in health-related quality of life were found over time for either group, while both groups showed statistically non-significant changes over time in pain disability. This suggests that neither treatment reduced the extent to which the chronic musculoskeletal pain interfered with the patients' lives [46, 50]. It is possible that the outbreak of COVID-19 and its societal consequences might have played a confounding role, as part of the data was collected during the outbreak. For instance, it is found in Germany that people who refrain themselves from going to public places show significantly lower health-related quality of life [56].

Although there is consensus that VNS affects pain processing, it remains unclear how this happens exactly [15, 16]. In the last decades, traditional VNS is gradually being replaced by safer t-VNS, but t-VNS is predominantly applied in the treatment of headache, epilepsy and depression disorders [17, 57]. Literature about the effect of electrical t-VNS on nociception is conflicting: while one study found an increase in pressure pain threshold and a decrease in reported pain ratings after an hour of transcutaneous stimulation of the auricular branch of the vagal nerve, another found that only most participants showed an increase in pain threshold after 30 minutes of similar stimulation [23, 24]. Moreover, some participants of the latter study even showed a decrease in pain threshold, indicating that t-VNS can result in both pro- and anti-nociceptive effect, depending on the individual. These latter results might explain why the low-frequency treatment did not outperform the high-frequency treatment in both short and long term, and why no long term effect was found for either treatment. Moreover, the result that the low-frequency treatment was a beneficial add-on in the treatment of depression [21], is in line with the idea that the low-frequency treatment leads to VNS.

Yet, it is more likely that the low-frequency treatment did not result in VNS. Although we have not measured (a proxy of) vagal activity, vibration-sensitive vagal afferents in the mesentery have never been found, and the indications that suggest that t-VNS might also be induced by vibratory stimulation are weak. This lack of evidence that HALF-MIS treatment actually leads to t-VNS is considered an important limitation of this study. Another limitation is that the anticipated number of 60 patients was not achieved: in March 2020, the study had to be terminated prematurely due to the COVID-19 outbreak. However, the data does not suggest that the hypotheses of this pilot study would have been confirmed if statistical power was higher. Feasibility was not examined in this pilot, which is retrospectively considered to be a limitation as well. A final limitation is that the COVID-19 outbreak and its societal consequences could have played a confounding role in this study. The isolating measures affect musculoskeletal pain as physical activity is generally drastically reduced, and perceived pain is often increased by anxiety and stress for instance, as pain can comprise cognitive, social and emotional components [58–60]. Since elderly are known to be at increased risk for an unfavorable course of illness [61], they are likely to be more affected by these consequences than any other group.

To our knowledge, this is the first study relying on the ability to excite abdominal vagal afferents transcutaneously with low-frequency vibrations in order to achieve pain reduction. However, changes in vagal activity have not been examined and it has not been proven that the mesenterial Pacinian corpuscles are innervated by the vagal nerve. The treatment proved to be safe, but both the low-frequency and the high-frequency treatments did not decrease CMP in elderly over time. A statistically significant but clinically insignificant immediate decrease in pain intensity was found for both treatments when comparing pre-treatment and post-treatment scores. Moreover, no effects were found on quality of life, pain disability or pain intensity over a longer period of time for both groups. The hypotheses that (I) the combination of vibrations administered to the abdomen and rhythmically aligned music would have a clinically meaningful analgesic effect in elderly with CMP and (II) the administration of low-frequency vibrations and music would be more effective than the same treatment, but with higher-frequency vibrations as a treatment for CMP in elderly are rejected. More research is needed in the treatment of CMP in elderly, as this population often has reduced treatment options due to comorbidities and poly-pharmacy. The treatment options that are left, often fail to achieve satisfactory pain relief. VNS has analgesic potential, but it remains unclear if the low-frequency treatment is able to induce VNS. Future research should focus on the relationship between VNS and pain processing and other possibilities to achieve transcutaneous VNS such as via the auricular branch of the vagal nerve.

## Supporting information

**S1 File. CONSORT 2010 checklist of information to include when reporting a randomised trial**∗**.**
(DOC)

**S2 File. Research protocol.**
(PDF)

**S3 File. Dataset.**
(SAV)

## Author Contributions

**Conceptualization:** Peter Michael L. Nielsen, André Paul Wolff, Remko Soer.

**Formal analysis:** Peter J. C. Stuijt, Remko Soer.

**Funding acquisition:** Remko Soer.

**Investigation:** Thom A. H. Eshuis, Peter J. C. Stuijt, Peter Michael L. Nielsen.

**Methodology:** Thom A. H. Eshuis, Peter Michael L. Nielsen, André Paul Wolff, Remko Soer.

**Project administration:** Thom A. H. Eshuis, Peter J. C. Stuijt, Remko Soer.

**Resources:** Thom A. H. Eshuis, Remko Soer.

**Supervision:** Hans Timmerman, André Paul Wolff, Remko Soer.

**Validation:** Hans Timmerman, Remko Soer.

**Visualization:** Thom A. H. Eshuis, Peter J. C. Stuijt.

**Writing – original draft:** Peter J. C. Stuijt.

**Writing – review & editing:** Thom A. H. Eshuis, Hans Timmerman, André Paul Wolff, Remko Soer.

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
