## [Decision Letter · Decision Letter 0]

11 Aug 2021

PONE-D-21-20572

The effect of High Amplitude Low Frequency – Music Impulse Stimulation on chronic musculoskeletal pain in elderly: a multicenter randomized controlled pilot study.

PLOS ONE

Dear Dr. Eshuis,

Thank you for submitting your manuscript to PLOS ONE. After careful consideration, we feel that it has merit but does not fully meet PLOS ONE’s publication criteria as it currently stands. Therefore, we invite you to submit a revised version of the manuscript that addresses the points raised during the review process.

We look forward to receiving your revised manuscript.

Kind regards,

Walid Kamal Abdelbasset, Ph.D.

Academic Editor

PLOS ONE

Journal Requirements:

“I have read the journal's policy and the authors of this manuscript have the following competing interests:

PMN is registered inventor of the HALF-MIS technology. As a consequence, PMN did not participate in statistics processing or discussion of the results. For the remaining authors none were declared.”

4. We note that Figure 2 in your submission contain copyrighted images. All PLOS content is published under the Creative Commons Attribution License (CC BY 4.0), which means that the manuscript, images, and Supporting Information files will be freely available online, and any third party is permitted to access, download, copy, distribute, and use these materials in any way, even commercially, with proper attribution. For more information, see our copyright guidelines: http://journals.plos.org/plosone/s/licenses-and-copyright.

Reviewers' comments:

Reviewer's Responses to Questions

**Comments to the Author**

1. Is the manuscript technically sound, and do the data support the conclusions?

Reviewer #1: Partly

Reviewer #2: No

Reviewer #3: Yes

2. Has the statistical analysis been performed appropriately and rigorously? 

Reviewer #1: No

Reviewer #2: I Don't Know

Reviewer #3: I Don't Know

3. Have the authors made all data underlying the findings in their manuscript fully available?

Reviewer #1: No

Reviewer #2: No

Reviewer #3: No

4. Is the manuscript presented in an intelligible fashion and written in standard English?

Reviewer #1: Yes

Reviewer #2: No

Reviewer #3: Yes

5. Review Comments to the Author

Reviewer #1: This is a very interesting randomised pilot study examining the effect of music and low frequency vibrations on chronic musculoskeletal pain in elderly patients. Well done to team for carrying out such important research.

The major comment is, this is a pilot study looking at efficacy and safety of an intervention. Which is misleading as generally speaking pilot studies are requisites to the initial step in exploring the feasibility of pre-specified approaches (e.g recruitment rate, retentions, adherence etc) intended for the larger study. Also the sample size does not suggest or reflect what the effect size is. Thus the making conclusions about clinically meaningful results misleading as we don’t know what effect size the study was designed to detect.

They are some comments worth mentioning for the authors attention.

1) Sample size section should come before statistical analysis section,

2) Line 220 “All analyses are performed according the intention-to-treat principle” can the authors explicitly define this. as CONSORT and Table 1 don't agree.

3) As this is a pilot randomised controlled trial, it’s not recommended to test baseline characteristics as any differences that may occur are simply due to chance and also due to randomisation you expect a balance between groups. i.e remove p-values from Table 1. i.e If analysis was based on ITT, why does Table 1 include =45 instead of the n=46 randomised individuals.

4) Table 1 – usually a table showing baseline characteristics by randomised group. Results stratified by centre should be omitted from this table. Also please indicate what summary statistic has been presented. i.e. Age(years) ?median [?IQR] do the same for other variables.

5) The authors have used non-parametric methods on parametric estimates.. If Wilcoxon has been used meaning data is believed to be skewed therefore this should be used to test if there is a difference in medians? Similar comment for use of Mann-Whitney tests.

6) Authors should make reference to Cohen’d effect size and not just state ‘effect’ size

7) In the statistical analysis, authors should mention, how missing data would be handled.

8) Was there an analysis plan and finalised prior to final analysis?

9) Within group differences have been presented in Table 2, has multiple testing been accounted for? And not sure this is worth presenting as a linear mixed effect model has been fitted.

10) Where the variables chosen to be included in the model decided as priori?

11) Table 3 should present the unadjusted and adjusted group estimates i.e. for T1 and T2 – together with 95%CIs and p-values

Reviewer #2: Title

Kindly frame title such that it is accurate, informative, descriptive, succinct, simple and specific.

Introduction

Totally need editing and rewrite and it is too long

Methods:

Kindly focus on three basic elements of methods section.

How the study was designed?

How the study was carried out?

How the data were analyzed?

Eligibility criteria for participants not clear and Provide sufficient details of interventions of each group to allow replication

Statistics

Completely Not clear –please rewrite again and discuss which test you used and why also all tables need rearrange

Outcomes and estimation need to be explained well

Discussion:

needs to be as per well defined objectives

Describe sources of potential bias and imprecision

It has to be framed in such a way that readers are able to have good understanding of the current evidences and rationale of the paper

Reviewer #3: Dear Authors,

Kindly consider the following amendments or respond to them

Abstract:

Conclusion: line 53 the phrase “assumed” does not reflect a conclusive summary… recommend to change to “we believe”

Line 55: trial registration should be removed to the design and methodology section

Introduction

Line 61: you mentioned LBP, neck pain and OA and just gave 2 prevalence (4th and 13th)

Line 108: you need to rely on more than 1 reference (38) to support the efficacy of low frequency vibrations

The rationale or research gaps still needs to be more clear

Materials and methods:

Design:

Line 125: As the study was conducted between 125 August 2019 and March 2020, are there any reasons to state publishing your results after 2 years?

Line 126: although it is stated that San Martino Hospital (Genova, Italy) participated in the study settings but not of the author affiliations represent it!

Patients:

Line 132: what about rheumatic arthritic conditions? Were they excluded?

How was the analgesic effect of medication controlled? Especially if patients were using different medications?

Procedure:

Line 152: full term before acronyms in the first statement

Kindly list who provided the therapy and his/her expertise

Line 180-182: reference for the time and frequency of treatment

Outcomes:

list validity and reliability of the outcome measures used

line 209: has the translation been validated? How?

Statistical analysis:

Line 220: why was the intent to treat used since it’s a pilot study?

Results:

Table 1: kindly add an astrex “*” to the significant value “ weight”

Table 2: p values missing for NRS

Line 304: full term of “AE”

Discussion:

Lines 345-349: are there no other justifications for the non statistical difference rather than the Covid?

The discussion is well written but still lacks consolidated justification of the findings

6. PLOS authors have the option to publish the peer review history of their article (what does this mean?). If published, this will include your full peer review and any attached files.

Reviewer #1: No

Reviewer #2: No

Reviewer #3: **Yes: **Tamer M. Shousha

---

## [Author Response · Author response to Decision Letter 0]

11 Oct 2021

1. Is the manuscript technically sound, and do the data support the conclusions?

Reviewer #1: Partly

Reviewer #2: No

Reviewer #3: Yes

Answer: Throughout the conduct of the study, we did face some obstacles that led to certain limitations (e.g. the COVID pandemic and the exclusion of the Italian site). Despite these limitations, we are convinced to have produced a technically sound manuscript with conclusions that are supported by the data.

2. Has the statistical analysis been performed appropriately and rigorously? 

Reviewer #1: No

Reviewer #2: I Don't Know

Reviewer #3: I Don’t Know

Answer: We have improved the readability and clearity of our “Statistical analysis” – paragraph, and hope this allows the reviewers to judge our analyses more easily.

3. Have the authors made all data underlying the findings in their manuscript fully available?

Reviewer #1: No

Reviewer #2: No

Reviewer #3: No

Answer: We have now added the minimal data set that was used for this study.

4. Is the manuscript presented in an intelligible fashion and written in standard English?

Reviewer #1: Yes

Reviewer #2: No

Reviewer #3: Yes

Answer: We have thoroughly inspected the language of our manuscript, and are pleased to see that no specific grammar, spelling or fashion errors are specified by the reviewers.

5. Review Comments to the Author

Reviewer #1: This is a very interesting randomised pilot study examining the effect of music and low frequency vibrations on chronic musculoskeletal pain in elderly patients. Well done to team for carrying out such important research.

The major comment is, this is a pilot study looking at efficacy and safety of an intervention. Which is misleading as generally speaking pilot studies are requisites to the initial step in exploring the feasibility of pre-specified approaches (e.g recruitment rate, retentions, adherence etc) intended for the larger study. Also the sample size does not suggest or reflect what the effect size is. Thus the making conclusions about clinically meaningful results misleading as we don’t know what effect size the study was designed to detect.

Answer: We thank the reviewer for the kind words on the conduct of our study. We understand that not elaborately examining the feasibility is a limitation, which we now addressed in the limitation section in lines 448-449: “Feasibility was not examined in this pilot, which is retrospectively considered to be a limitation as well”. Yet, we think it is correct to refer to this study as a pilot study, since we could not make an estimate on the effect size before the start of the trial, as no comparable data was available, as mentioned in lines 237-238: “Since this was the first trial investigating the effect of HALF-MIS on CMP in elderly, there was no published or unpublished pilot data available to base a sample size on.”. 

Regarding the strength of our drawn conclusions, we agree that we might have been too strong when stating that the achieved pain relief was generally not clinically meaningful, and that we should be more tentative in our conclusions. Please take note of the following adaptions that have been made to be more tentative in our conclusion and avoid “misleading conclusions” (to cite Reviewer #1): 

- In the abstract, we specified that the effect of either treatment over time was clinically not meaningful (instead of initially stating that the treatment was generally clinically not meaningful (see lines 55-57: “However, over time, neither the low-frequency treatment group nor the high-frequency treatment group provided clinically meaningful pain relief.”)

- In the discussion a more tentative conclusion is drawn (see lines 383-387: “It was also expected that the combination of vibrations administered to the abdomen and rhythmically aligned music would have a clinically meaningful analgesic effect in elderly with CMP, but as no time effect was found either, it seems that both treatments do not decrease CMP in elderly in the long run.”)

- In the discussion, we used different wording when interpreting our results (see lines 404-409: “While the effect sizes are moderate (r = -0.49) for the low-frequency and large (r = -0.50) for the high-frequency group, these changes are not always significant from the patient’s perspective; as the median drop in mean pain intensity score per patient is 19% for the low-frequency group and 16% for the high-frequency group, most patients experience only partial temporary relief of their moderate to severe musculoskeletal pain immediately after treatment.”)

- In the discussion, the word “irrelevant” was replaced by “insignificant” (see lines 460-462: “A statistically significant but clinically insignificant immediate decrease in pain intensity was found for both treatments when comparing pre-treatment and post-treatment scores.” )

- In general, the results regarding the safety of the treatment have become more pronounced (see for instance lines 46-47: “Primary outcomes were safety (Serious Adverse Events) and pain intensity, measured at baseline, after the last treatment and at six weeks follow-up.”, lines 49-51: “. After 344 treatments, 1 Adverse Event was found related to the intervention, while 13 Adverse Events were possibly related.”, lines 54-55: “The combination of music and abdominally administered vibrations was found to be safe and well tolerated by the elderly patients.”, lines 320-325: “The treatments appeared well received and safe. No Serious Adverse Events that were possibly related to the intervention were reported. One non-serious Adverse Event was reported that was related to the treatments as one subject perceived low back pain caused by poor posture while treatment was administered. 13 other non-serious Adverse Events might have been related, of which short episodes of respectively dizziness/vertiginous (n = 5) and headache (n = 2) and tiredness (n = 2) have been reported more than once.”, and lines 387-389: “The expectation that the treatment would be safe and well tolerated by the patients was confirmed, judging by both the quantity and quality of reported (Serious) Adverse Events that were possible related to the intervention.”)

They are some comments worth mentioning for the authors attention.

1) Sample size section should come before statistical analysis section,

Answer: Thank you for bringing this to our attention. We have changed this accordingly.

2) Line 220 “All analyses are performed according the intention-to-treat principle” can the authors explicitly define this. as CONSORT and Table 1 don’t agree.

Answer: There was indeed a mistake in the CONSORT Flow diagram, which we have corrected. Moreover, we have now explicitly defined how all analyses have been performed regarding the intention-to-treat analysis (see lines 242-244: “All analyses are performed according the intention-to-treat principle to avoid potential exclusion bias, meaning that patients who started but did not finish the intervention were included in the analyses (53).”). To clarify the exact number of patients that have been included in the analyses; 46 patients were randomized in the trial, but 1 of these was not seen for baseline measurement or had any treatment. We have explained this now in lines 280-282: “A total of 46 patients have been included and randomized (see Fig 1 and Table 1). However, one patient had not started the intervention when the study was terminated due to the COVID-19 outbreak, and was therefore excluded from the analyses.”.

3) As this is a pilot randomised controlled trial, it’s not recommended to test baseline characteristics as any differences that may occur are simply due to chance and also due to randomisation you expect a balance between groups. i.e remove p-values from Table 1. i.e If analysis was based on ITT, why does Table 1 include =45 instead of the n=46 randomised individuals.

Answer: We agree that any differences between groups are not expected. However, any differences that simply occur by chance, could potentially influence the effectiveness of the treatment, and lead to biased results. For the sake of transparency, we have kept the p-values in Table 1. For the ITT, please see our answer under comment #2, above.

4) Table 1 – usually a table showing baseline characteristics by randomised group. Results stratified by centre should be omitted from this table. Also please indicate what summary statistic has been presented. i.e. Age(years) ?median [?IQR] do the same for other variables.

Answer: We agree with the reviewer, and removed the centre groups from the Table. Additionally, we indicated which summary statistic was used per variable. 

5) The authors have used non-parametric methods on parametric estimates.. If Wilcoxon has been used meaning data is believed to be skewed therefore this should be used to test if there is a difference in medians? Similar comment for use of Mann-Whitney tests.

Answer: We believe this is a misunderstanding, probably caused by our wording, as both Wilcoxon and Mann-Whitney tests have been used on non-parametric estimates (i.e. medians, not means). What we have done is calculate e.g. the mean NRS score per patient before each treatment, and also calculate the mean NRS score per patient after each treatment, so that you get two variables which are scored per patient, called “mean NRS score pre-treatment” and “mean NRS score post-treatment”. Then, we compared these by means of a Wilcoxon test (using the medians of the [mean NRS score per patient]), since these two variables were not meeting the assumptions required for parametric testing etc. 

We hope to prevent future misunderstandings, by replacing the word “mean” by “average” in lines 257-261 (“To assess the immediate pain relief within treatment groups, the average NRS score per patient before the delivered treatments was compared to the average NRS score per patient after the delivered treatments for both groups using Wilcoxon signed-rank test. The average difference between pre- and post-treatment NRS per patient between treatment groups were compared by means of Mann-Whitney test.”) and lines 356-365 (“With regard to the immediate effects of both treatments, related-samples Wilcoxon Signed Rank Tests showed for both treatment groups that the average NRS score per patient after treatment was significantly lower than the average NRS per patient before treatment: for the low-frequency group, the average NRS score after treatment (Mdn = 3.9, IQR = 2.3-5.0) was lower than the average NRS score before treatment (Mdn = 4.8, IQR = 3.4-5.4), p < 0.01, r = -0.49, while for the high-frequency group, the average NRS score after treatment (Mdn = 3.6, IQR = 2.5-5.3) was lower than the average NRS score before treatment (Mdn = 4.3, IQR = 3.3-6.6), p < 0.01, r = -0.50. The average difference per patient between pre- and post-treatment NRS scores from the low-frequency group (Mdn = 0.8, IQR = 0.4–1.5) did not differ significantly from those of the high-frequency group (Mdn = 0.6, IQR = 0.1–1.3), U = 235.00, z = -0.41, p = 0.68, r = -0.06.”), to distinguish between the meaning related to its summarizing nature (to which we want to refer) and the meaning related to a statistical method.

6) Authors should make reference to Cohen’d effect size and not just state ‘effect’ size

Answer: We did not use Cohen’s d, but r as an effect size. We believe that r is a more appropriate than Cohen’s d in case of non-parametric testing, as the latter assumes normal distribution and is based on means, and should therefore only be used as a parametric measure of effect size.

7) In the statistical analysis, authors should mention, how missing data would be handled.

Answer: We agree, and have added information to our methods section (see lines 255-256: “As multilevel models do not require completeness of data, missing data was not imputed.” and lines 261-265: “Exclusion bias and missing data bias have been reduced by first taking the mean of all pre-treatment and post-treatment NRS scores for the Wilcoxon signed-rank test and mean difference per patient between each pre- and post-treatment NRS scores for the Mann-Whitney test, before conducting the tests mentioned.”). We did not impute missing data.

8) Was there an analysis plan and finalised prior to final analysis?

Answer: Due to the exploratory nature of this study, we did not a priori know which exact choices in statistics would be made. There was a global analysis plan, which was stated in the study protocol and approved by the METC.

9) Within group differences have been presented in Table 2, has multiple testing been accounted for? And not sure this is worth presenting as a linear mixed effect model has been fitted.

Answer: We added to the paragraph “Statistical analysis” (lines 271-273, “To control for familywise error in the comparison of within-group differences for the secondary outcomes, Bonferroni correction was used and the two-sided level of statistical significance was set to 0.0125.”). The multilevel linear model was only used for the prediction of pain intensity scores, and our secondary outcome measures have not been included in the model. Therefore, we believe presenting the within-group differences of our secondary outcomes is of added value. 

10) Where the variables chosen to be included in the model decided as priori?

Answer: Group and Time and Center effects were included a priori in the study protocol.

11) Table 3 should present the unadjusted and adjusted group estimates i.e. for T1 and T2 – together with 95%CIs and p-values

Answer: Also in line with the feedback from Reviewer#2, we have rearranged the tables. Table 2 now describes pain intensity and analgesic use for T0, T1 and T2. As both outcomes have been included and tested in the multilevel linear model, no test statistics are given in Table 2. In Table 3, the secondary outcomes are both described and tested (with p-values provided), as these are not included in the model. In Table 4, the parameter estimates of the model are given, with 95% CI’s.

Reviewer #2: Title

Kindly frame title such that it is accurate, informative, descriptive, succinct, simple and specific.

Answer: We have changed the title to “Music and low-frequency vibrations for treatment of chronic musculoskeletal pain in elderly: a pilot study”.

Introduction

Totally need editing and rewrite and it is too long

Answer: We have critically looked at our introduction, and omitted any unnecessary detailed information. As stated in the discussion, the indications that suggest that t-VNS might also be induced by vibratory stimulation are weak. We aimed to provide an comprehensive background and rationale for this study, which is also intelligible for readers that are no expert in the fields of pain medicine, vagal nerve stimulation or vibro-acoustic therapy.

Methods:

Kindly focus on three basic elements of methods section.

How the study was designed?

How the study was carried out?

How the data were analyzed?

Answer: We adhered to the CONSORT 2010 checklist, which is required by PLOS ONE when reporting a clinical trial. We believe that we have reported the methods of this study concisely.

Eligibility criteria for participants not clear 

Answer: We have improved the readability of the eligibility criteria (see lines 141-151: “Patients were included when at least 65 years, were eligible if suffering from the symptoms of pain for at least≥ 3 months, with a minimum intensity of Numeric (Pain) Rating Scale (NRS) of 4. Moreover, the pain had to be the result of a condition diagnosed as a musculoskeletal disease listed in the International Classification of Diseases-10 of 2014 under M00-M99 (44). Patients participating in other (experimental) clinical studies, undergoing any kind of music therapy as a current treatment for their pain, suffering from active or untreated comorbidities (including moderate to severe depression), having pain related to malignancies, showing signs of the pain syndrome being exclusively or predominantly neurogenic during physical examination or being diagnosed with prolapsus disci intervertebralis with myelopathy/radiculopathies (listed as M50.0, M50.1, M51.0 and M51.1 in the International Classification of Diseases-10 of 2014) were excluded (44).”).

and Provide sufficient details of interventions of each group to allow replication

Answer: We agree on the reviewer and we inserted a paragraph entitled “Interventions” (starting from line 177). This should now enable replication of this study. 

Statistics

Completely Not clear –please rewrite again and discuss which test you used and why also all tables need rearrange

Answer: We have elaborated the description of our statistics (please see lines 242-244: “All analyses are performed according the intention-to-treat principle to avoid potential exclusion bias, meaning that patients who started but did not finish the intervention were included in the analyses (53).”, lines 261-265: “Exclusion bias and missing data bias have been reduced by first taking the mean of all pre-treatment and post-treatment NRS scores for the Wilcoxon signed-rank test and mean difference per patient between each pre- and post-treatment NRS scores for the Mann-Whitney test, before conducting the tests mentioned.” and lines 271-274: “To control for familywise error in the comparison of within-group differences for the secondary outcomes, Bonferroni correction was used and the two-sided level of statistical significance was set to 0.0125. All other statistical analyses were conducted with a two-sided 0.05 level of statistical significance, using SPSS statistics (version 23.0; IBM Corp. Armonk, NY).”. Also in line with the feedback from Reviewer#1 comment #11, we have rearranged the tables. Table 2 now describes pain intensity and analgesic use for T0, T1 and T2. As both outcomes have been included and tested in the multilevel linear model, no test statistics are given in Table 2. In Table 3, the secondary outcomes are both described and tested (with p-values provided), as these are not included in the model. In Table 4, the parameter estimates of the model are given, with 95% CI’s.

Outcomes and estimation need to be explained well

Answer: We shifted the focus slightly to safety (besides the efficacy, see the last point of our answer to the general comment of Reviewer#1 above). Moreover, we have also added comments about the reliability of our outcome measures, in line with the feedback of Reviewer#3 (see for instance lines 205-207: “The NRS score provides a valid and reliable measure for pain intensity for this study population, and measures pain on a 11-point scale ranging from 0 (no pain) to 10 (the worst pain imaginable) (45).”, lines 215-216 “The EQ-5D-3L has been proven a valid tool for assessing quality of life in Danish, Dutch and Italian (46) and .is found to be reliable when used in a Dutch elderly population (47).”and lines 222-223 “The PDI has been proven a valid and reliable tool for measuring disability associated with pain in Dutch, and consists of 7 items which are scored on a 11-point scale (50).”).

Discussion:

needs to be as per well defined objectives

Answer: We have edited our discussion; in the discussion, we now refer to the hypotheses that were composed in the introduction (see lines 378-389: “It was hypothesized that the combination of music and vibrations with frequencies between 20-100 Hz would be more effective in relieving CMP in elderly than the combination of music and vibrations with higher frequencies (200-300 Hz). However, there was no statistically significant interaction effect of group by time in the multilevel linear model predicting pain intensity, meaning that the low-frequency treatment was not more effective than the high-frequency treatment when comparing pain intensity at baseline, after the last treatment and at follow-up, respectively. It was also expected that the combination of vibrations administered to the abdomen and rhythmically aligned music would have a clinically meaningful analgesic effect in elderly with CMP, but, as no time effect was found either, it seems that both treatments do not decrease CMP in elderly in the long run.. The expectation that the treatment would be safe and well tolerated by the patients was confirmed, judging by both the quantity and quality of reported (Serious) Adverse Events that were possibly related to the intervention.”), and concisely reflect upon these by interpreting our results (e.g. lines 402-409: “However, both the low-frequency and the high-frequency treatments showed a statistically significant decrease in pain intensity when pain intensity scores before and after the treatment are compared. While the effect sizes are moderate (r = -0.49) for the low-frequency and large (r = -0.50) for the high-frequency group, these changes are not always significant from the patient’s perspective; as the median drop in mean pain intensity score per patient is 19% for the low-frequency group and 16% for the high-frequency group, most patients experience only partial temporary relief of their moderate to severe musculoskeletal pain immediately after treatment.” and comparing these with existing literature (see e.g. lines 409-412: “Still, the immediate effects found in this study are larger than immediate-term effects of oral and transdermal opioid analgesics on pain intensity in elderly with CMP found by a systematic review, which showed a standardized mean difference of 0.27 in a sample of 4998 patients by combining 16 studies (55).”).

Describe sources of potential bias and imprecision. It has to be framed in such a way that readers are able to have good understanding of the current evidences and rationale of the paper

Answer: we described potential biases and limitations in line 441-455: “Yet, it is more likely that the low-frequency treatment did not result in VNS. Although we have not measured (a proxy of) vagal activity, vibration-sensitive vagal afferents in mesentery have never been found, and the indications that suggest that t-VNS might also be induced by vibratory stimulation are weak. This lack of evidence that HALF-MIS treatment actually leads to t-VNS is considered an important limitation of this study. Another limitation is that the anticipated number of 60 patients was not achieved: in March 2020, the study had to be terminated prematurely due to the COVID-19 outbreak. However, the data does not suggest that the hypotheses of this pilot study would have been confirmed if statistical power was higher. Feasibility was not examined in this pilot, which is retrospectively considered to be a limitation as well. A final limitation is that it is likely that the COVID-19 outbreak and its societal consequences could have played a confounding role in this study. The isolating measures affect musculoskeletal pain as physical activity is generally drastically reduced, and perceived pain is often increased by anxiety and stress for instance, as pain can comprise cognitive, social and emotional components (58, 59, 60). Since elderly are known to be at increased risk for an unfavorable course of illness (61), they are likely to be more affected by these consequences than any other group.”. Please note that this includes the limitation of us not addressing the feasibility of this study.

Reviewer #3: Dear Authors,

Kindly consider the following amendments or respond to them

Abstract:

Conclusion: line 53 the phrase “assumed” does not reflect a conclusive summary… recommend to change to “we believe”

Answer: We fully agree that “assumed” does not reflect a conclusive summary, and changed this to: “There is no evidence that the low-frequency treatment elicited vagal nerve stimulation” (lines 57-58). In our opinion, this is better than “we believe”, as we feel this way of expression still lacks conclusive properties.

Line 55: trial registration should be removed to the design and methodology section

Answer: Although we fully agree that trial registration does indeed belong to the design and methodology section, PLOS ONE also requires authors to state the trial registration in the abstract, in case of clinical trials. Therefore, we did not edit this part of the abstract.

Introduction

Line 61: you mentioned LBP, neck pain and OA and just gave 2 prevalence (4th and 13th)

Answer: We provided three causes of years lived with disability, namely: Leading (I) 4th (II) and 13th (III).

Line 108: you need to rely on more than 1 reference (38) to support the efficacy of low frequency vibrations

Answer: We have added a reference to a paper which supports the efficacy of low-frequency vibrations (line 112). We hope this suffices.

The rationale or research gaps still needs to be more clear

Answer: We completely agree that there are considerable research gaps in this field of research. Indeed, there has not been any anatomical, nor physiological studies directly showing connections between sensors in the abdomen and the vagal nerve. In the introduction, we aimed to fully expose these research gaps, while remaining concise and avoiding being exhaustive (with regard to Reviewer#2, the comment about the introduction being too long). Moreover, we address these research gaps as important limitations (see lines 441-445: “Yet, it is more likely that the low-frequency treatment did not result in VNS. Although we have not measured (a proxy of) vagal activity, vibration-sensitive vagal afferents in the mesentery have never been found, and the indications that suggest that t-VNS might also be induced by vibratory stimulation are weak. This lack of evidence that HALF-MIS treatment actually leads to t-VNS is considered an important limitation of this study.”). In order to progress in this field, we conducted this pilot study.

Materials and methods:

Design:

Line 125: As the study was conducted between 125 August 2019 and March 2020, are there any reasons to state publishing your results after 2 years?`

Answer: When the study was prematurely terminated in March 2020, we initially awaited how the COVID pandemic would develop, as we were hoping for the trial to be continued. After some time and analyses, the study was terminated definitively. Yet, only one year has passed since the first submission in the meantime, and it has been submitted to two different journals previously.

Line 126: although it is stated that San Martino Hospital (Genova, Italy) participated in the study settings but not of the author affiliations represent it!

Answer: As the authors were not defined a priori, and after excluding the Italian partners, none of them met the ICMJE criteria for authors. 

Patients:

Line 132: what about rheumatic arthritic conditions? Were they excluded?

Answer: Since rheumatic arthritic conditions are listed under M05 and M06 in the ICD-10, these patients were not excluded a priori, but eligible if meeting the other inclusion criteria. 

How was the analgesic effect of medication controlled? Especially if patients were using different medications?

Answer: Since the intervention was in addition to care as usual, patients were allowed to continue, stop or start any pain medication. We controlled for analgesic use by means of keeping track of patients’ analgesic use throughout the trial, which was quantified using the Analgesic Intake Scale (lines 230-235: “Use of analgesics over time during intervention period was considered a possible confounder and was therefore also recorded at the screening visit. Moreover, changes in analgesic use were kept track of at each visit. Analgesics intake was then quantified using the Analgesics Intake Scale (52). This tool yields a score from 0 (no use of pain medication) to 8 (use of opioids in combination with benzodiazepines/antidepressant/anticonvulsants), and allows for comparing analgesic intake between patients by taking into account both type and quantity of medication used.”). Based on this, we concluded that there were no significant changes in medication intake during the trial. Moreover, the Analgesic Intake Scale was used in the multilevel linear model, but did not show to be a predictor of pain intensity. 

Procedure:

Line 152: full term before acronyms in the first statement

Answer: We have made sure the full term is listed first, before using acronyms for referral.

Kindly list who provided the therapy and his/her expertise

Answer: We have added that the treatments were provided by physicians (and [co-]authors) Eshuis and Nielsen (see lines 178-179: “Treatments were administered by physicians T.E. and P.N., in the Netherlands and Denmark, respectively.”).

Line 180-182: reference for the time and frequency of treatment

Answer: We have added a reference for the time and frequency of the treatment (see line 183). 

Outcomes:

list validity and reliability of the outcome measures used

Answer: We have added the reliability of the outcome measures, as validity was already given (see for instance lines 205-207: “The NRS score provides a valid and reliable measure for pain intensity for this study population, and measures pain on a 11-point scale ranging from 0 (no pain) to 10 (the worst pain imaginable) (45).”, lines 215-216 “The EQ-5D-3L has been proven a valid tool for assessing quality of life in Danish, Dutch and Italian (46) and .is found to be reliable when used in a Dutch elderly population (47).”and lines 222-223 “The PDI has been proven a valid and reliable tool for measuring disability associated with pain in Dutch, and consists of 7 items which are scored on a 11-point scale (50).”.

line 209: has the translation been validated? How?

Answer: The PDI has been validated in other languages including Dutch. We did not perform a formal validation for Danish or Italian, which is a limitation. Yet, we believe that within the scope of this study and its results (in which the PDI is only a secondary outcome), this limitation is not worth mentioning when compared to other limitations with bigger impact (e.g. the thin indications that suggest that t-VNS might also be induced by vibratory stimulation).

Statistical analysis:

Line 220: why was the intent to treat used since it’s a pilot study?

Answer: We believed that it would be valuable to apply the intention-to-treat principle and reduce potential exclusion bias, as applying the intention-to-treat principle yields the most generalizable results and expectations based on realistic adherence, for when the treatment was to be implemented in usual care. In the hypothetical case that adherence to the treatment was low, we would not have overestimated the effect of the treatment, which would happen when applying the per-protocol principle instead. 

Results:

Table 1: kindly add an astrex “*” to the significant value “ weight”

Answer: We agree that adding an asterix would have been appropriate to indicate the statistically significant difference in weight between both treatment groups. However, Reviewer#1 requested to omit the comparison of demographic characteristics between the two centers. Therefore, we have omitted these comparisons in Table 1.

Table 2: p values missing for NRS

Answer: In Table 2, only pain intensity and analgesic intake are described at T0, T1 and T2, which are tested in the multilevel linear model. Hence, there are no p-values in that table, as that would result double testing.

Line 304: full term of “AE”

Answer: We have replaced all AE abbreviations with the full term ‘adverse event’.

Discussion:

Lines 345-349: are there no other justifications for the non statistical difference rather than the Covid?

The discussion is well written but still lacks consolidated justification of the findings

Answer: We think the main reason for finding non-significant differences is because the low-frequency treatment might not elicit vagal nerve stimulation (see lines 441-445, “Yet, it is more likely that the low-frequency treatment did not result in VNS. Although we have not measured (a proxy of) vagal activity, vibration-sensitive vagal afferents in mesentery have never been found, and the indications that suggest that t-VNS might also be induced by vibratory stimulation are weak. This lack of evidence that HALF-MIS treatment actually leads to t-VNS is considered an important limitation of this study.”).

---

## [Decision Letter · Decision Letter 1]

19 Oct 2021

Music and low-frequency vibrations for the treatment of chronic musculoskeletal pain in elderly: a pilot study.

PONE-D-21-20572R1

Dear Dr. Eshuis,

We’re pleased to inform you that your manuscript has been judged scientifically suitable for publication and will be formally accepted for publication once it meets all outstanding technical requirements.

Kind regards,

Walid Kamal Abdelbasset, Ph.D.

Academic Editor

PLOS ONE

Additional Editor Comments (optional):

Reviewers' comments:

Reviewer's Responses to Questions

**Comments to the Author**

1. If the authors have adequately addressed your comments raised in a previous round of review and you feel that this manuscript is now acceptable for publication, you may indicate that here to bypass the “Comments to the Author” section, enter your conflict of interest statement in the “Confidential to Editor” section, and submit your "Accept" recommendation.

Reviewer #1: All comments have been addressed

Reviewer #2: All comments have been addressed

Reviewer #3: All comments have been addressed

2. Is the manuscript technically sound, and do the data support the conclusions?

Reviewer #1: (No Response)

Reviewer #2: Yes

Reviewer #3: Partly

3. Has the statistical analysis been performed appropriately and rigorously? 

Reviewer #1: Yes

Reviewer #2: Yes

Reviewer #3: I Don't Know

4. Have the authors made all data underlying the findings in their manuscript fully available?

Reviewer #1: No

Reviewer #2: Yes

Reviewer #3: Yes

5. Is the manuscript presented in an intelligible fashion and written in standard English?

Reviewer #1: Yes

Reviewer #2: Yes

Reviewer #3: Yes

6. Review Comments to the Author

Reviewer #1: (No Response)

Reviewer #2: thanks alot or your response

Reviewer #3: Most of the comments have been addressed.. Thank you for the response

7. PLOS authors have the option to publish the peer review history of their article (what does this mean?). If published, this will include your full peer review and any attached files.

Reviewer #1: No

Reviewer #2: No

Reviewer #3: No

---

## [Editor Report · Acceptance letter]

22 Oct 2021

PONE-D-21-20572R1 

Music and low-frequency vibrations for the treatment of chronic musculoskeletal pain in elderly: a pilot study. 

Dear Dr. Eshuis:

I'm pleased to inform you that your manuscript has been deemed suitable for publication in PLOS ONE. Congratulations! Your manuscript is now with our production department. 

Kind regards, 

on behalf of

Dr. Walid Kamal Abdelbasset 

Academic Editor

PLOS ONE